# Application of Composite Flour from Indonesian Local Tubers in Gluten-Free Pancakes

**DOI:** 10.3390/foods12091892

**Published:** 2023-05-04

**Authors:** Herlina Marta, Christine Febiola, Yana Cahyana, Heni Radiani Arifin, Fetriyuna Fetriyuna, Dewi Sondari

**Affiliations:** 1Department of Food Technology, Faculty of Agro-Industrial Technology, Universitas Padjadjaran, Bandung 45363, Indonesia; christine19002@mail.unpad.ac.id (C.F.); y.cahyana@unpad.ac.id (Y.C.); heni.radiani@unpad.ac.id (H.R.A.); fetriyuna@unpad.ac.id (F.F.); 2Research Center for Biomass and Bioproducts, Cibinong Science Center, National Researchand Innovation Agency, West Java, Cibinong 16911, Indonesia; dewi004@brin.go.id

**Keywords:** composite flour, local tubers, gluten-free, pancakes

## Abstract

Pancakes are fast food snacks that are generally made with wheat flour as the basic ingredients, which is an imported commodity and detrimental for people who are allergic to gluten. To reduce the use of wheat, alternative raw materials derived from local commodities are used, such as modified cassava flour (mocaf), arrowroot flour, and suweg flour. The experiment was carried out by mixing mocaf flour, arrowroot flour, and suweg flour to produce composite flour with a ratio of 70:15:15 (CF1), 70:20:10 (CF2), and 70:20:5 (CF3). The result showed that the ratio of mocaf flour, arrowroot flour, and suweg flour had a significant effect on pasting temperature, peak viscosity, hold viscosity, breakdown viscosity, setback, L*, a*, hue, whiteness, ∆E, as well as swelling volume and solubility on the characteristics of the composite flour. There was also a significant effect on the texture characteristics of hardness, adhesiveness, chewiness, color characteristics L*, a*, whiteness, ∆E, and flavor preference for the gluten-free pancake products. The best formulation to produce pancakes that have characteristics similar to wheat flour-based pancakes was 70% mocaf flour, 15% arrowroot flour, and 15% suweg flour.

## 1. Introduction

Flour is a raw material that plays an important role in Indonesian food and is widely used in food processing. The flour that is most often used in Indonesia is wheat flour, which is derived from wheat. Until now, Indonesia has needed to import wheat to fulfill high domestic demand. Importing wheat is also necessary because the wheat plant can only grow in subtropical regions. To reduce dependence on wheat imports, the price of which is continuously rising, it is necessary for the food sector to diversify.

Another reason that food diversification is necessary is the gluten content of wheat flour. Gluten is a protein that can be found in wheat [1]. Gluten has a role in the characteristics and stickiness of a dough and is responsible for cohesiveness, viscosity, and elasticity, as well as affecting the water absorption capacity of the dough [2]. However, the presence of gluten in foodstuffs can cause disturbances in some people, including those with celiac disease. Celiac disease is a genetic digestive disorder where the presence of gluten protein causes problems with the absorption of specific proteins in other food products [3]. Celiac disease affects as much as 1% of the human population, and the percentage does not vary based on age or race [4]. The presence of gluten in foodstuffs can also cause other disorders, such as non-celiac gluten sensitivity, dermatitis herpetiformis, gluten ataxia, and wheat allergy [5]. Therefore, alternative flours are needed to replace wheat flour.

Tubers are a source of carbohydrates that can be processed into alternative flours. Tuber flour has some advantages, such as being gluten-free and having raw materials that are available in Indonesia. However, the absence of gluten content in flour also has drawbacks, as products made without it have poorer characteristics than those cooked with it. Modifying the characteristics of non-gluten flours used is necessary to improve the products. Several previous studies have reported that starch/flour modification treatment can improve the flour’s characteristics and expand its application in the food industry [6,7,8,9]. A previous study also reported that modifications can improve the nutritional value of starch/flour [10,11]. For this reason, modifications were made to local tuber flour, namely cassava flour, using the fermentation method. The lack of characteristics of modified cassava flour (mocaf) can also be addressed by making composite flour, where mocaf flour is combined with other local tuber flours, such as arrowroot flour and suweg flour. The purpose of making composite flour is to improve the characteristics of the flour produced, as mixing the flours causes complementary properties.

One application of composite flour is an ingredient in gluten-free pancake production. Pancakes are fast food snacks with a flat, round shape, and are generally made with wheat flour as a basic ingredients. Good quality pancakes have perfect swelling power, and dough must always be fresh to meet good quality standards. Pancakes are generally consumed as an easy and quick breakfast to make at home; dough is made by mixing flour with milk, eggs, sugar, and baking powder, and is then fried on a pan [12]. Despite growing demand for them, most gluten-free products have poor sensory qualities [13,14]. Various studies on the formulation of gluten free pancakes combined jasmine and sanyod rice flour [3] as well as rice and sweet potato flour [15]. Based on research conducted by Shih et al. [15], making gluten-free pancakes with 100% rice flour as a base results in an unfavorable texture due to increased hardness when compared to pancakes made from wheat flour. Pancakes with a composition of 50% rice flour and 50% corn flour also produce unfavorable characteristics, including a brittle texture and a thick yellow color [16]. The poor results obtained can be caused by the absence of gluten content. The rheology of the dough and the finished product were both significantly impacted by the absence of gluten. Compared to the wheat dough, the gluten-free dough has poorer cohesion and elasticity [17,18,19,20]. In general, many deficiencies have been summarized in the characteristics of the gluten-free products, and there are a variety of interesting improvement approaches. This can be seen in the amount of research conducted regarding the manufacture of gluten-free flour, which is then used for various products [21,22,23,24,25,26,27]. However, no previous research has examined gluten-free composite flour made by mixing mocaf flour, arrowroot tuber flour, and suweg flour. Development of composite gluten-free flour can reduce dependence on wheat flour, which is rich in gluten, and reducing imports. To produce pancakes that meet the functional characteristics of flour while also having good characteristics and being liked by the panelists, it is necessary to find the appropriate ratio between mocaf flour, arrowroot tuber flour, and suweg flour.

## 2. Materials and Methods

### 2.1. Materials

The main ingredients used in this study were commercial wheat flour (Segitiga Biru by Bogasari Flour Mills from Indofood Sukses Makmur, Jakarta, Indonesia), commercial mocaf flour (Ladang Lima by Agung Bumi Agro, Surabaya, Indonesia), 7–8-month-old arrowroot tuber, and 10–12-month-old suweg tuber. Arrowroot tuber was obtained from Ciamis, Indonesia, and suweg tuber was obtained from a local farmer in Madiun, Indonesia.

### 2.2. Arrowroot Flour Preparation

Arrowroot flour was prepared according to Marta et al. [28] with a slight modification. The arrowroot tubers were washed and peeled, then reduced in size using a food processor into pieces with a thickness of 2–5 mm. The tuber slices were soaked in water to prevent enzymatic browning, then drained and dried in a cabinet oven at 60 °C for 24 h to reduce the water content. The dried tubers were milled using a miller machine and then sieved using a 100-mesh sieve. Sifted arrowroot flour was packed in a tight plastic zip top bag with silica gel to prevent an increase in water content in the flour.

### 2.3. Suweg Flour Preparation

Suweg flour was prepared according to Marta et al. [28] with a slight modification. The suweg tubers were washed and peeled, then reduced in size using a food processor to pieces with a thickness of 2–5 mm. During the cutting process, the tuber slices were soaked in water, and all parts were submerged to prevent enzymatic browning. The tuber slices were drained and dried in a cabinet oven at 60 °C for 24 h. The dried suweg tubers were milled using a miller machine, then sieved using a 100-mesh sieve. Sifted suweg flour was packed in a tight plastic zip top with silica gel to prevent an increase in the water content.

### 2.4. Pancake Preparation

Mocaf flour (MF), arrowroot flour (AF), and suweg flour (SF) were mixed and sieved to make a homogeneous composite flour (CF). For CF1, the ratio between flour was 70% mocaf flour, 15% arrowroot tuber flour, and 15% suweg flour; for CF2, 70% mocaf flour, 20% arrowroot tuber flour, and 10% suweg flour; and for CF3, 70% mocaf flour, 25% arrowroot tuber flour, and 5% suweg flour. Pancakes made from 100% wheat flour (WF) were used as controls. Each formulation (WF, CF1, CF2, and CF3) was combined with additional ingredients for making pancake products: 90 g of powdered sugar (Rose Brand by Adi Karya Gemilang), 9 g of baking powder (Koepoe Koepoe by Gunacipta Multirasa), and 26 g of powdered milk (Dancow by Nestle) [3], then mixed with 170 mL water and 1 beaten egg (±50 g). Formulation was mixed until the dough became homogeneous. Next, the dough was cooked with a diameter of ± 8 cm in a pan over low heat for ± 3 min [3]. Pancake formulations are presented in Table 1.

### 2.5. Proximate and Crude Fiber Composition Analysis of Flours

The moisture, ash, total protein, fat, and total crude fiber content were determined using standard [29] methods.

### 2.6. Pasting Properties (RVA Analysis) of Flours

The pasting properties of the flour samples were determined using a Rapid Visco Analyzer (RVA-SM2, Warriewood, Australia). In an RVA tube, 2.8 g of flour samples were added with 25 mL of aquadest (distilling water that has been cleaned of impurities and purified in a lab). The temperature was originally held at 50 °C for 1 min, then increased from 50 to 95 °C at a rate of 6 °C/min, held at 95 °C for 5 min, and then decreased to 50 °C at a rate of 6 °C/min. After that, the gel was preserved at 50 °C for 5 min.

### 2.7. Color Evaluation of Flours and Pancakes

The color scale for L*, a*, and b* of flour was measured using a Spectrophotometer CM-5 (Konica Minolta Co., Osaka, Japan) with Spectra Magic software. The color measurement includes L* (lightness, 0 = black/100 = white), a* (+a* = redness/−a* = greenness), b* (+b* = yellowness/−a* = blueness), and hue. The calibration was performed with a zero-calibration plate (CM-A124) and a white calibration plate (CM-A120) using a large target mask (CM-A203).

### 2.8. Water Absorption Capacity (WAC) Measurement of Flour

The water absorption capacity of the flour samples was determined according to Marta et al. [9]. A centrifuge tube containing 1 gram of flour was filled with 10 mL of water, allowed to remain at room temperature (26 ± 2 °C) for 1 h, and then centrifuged at 3500× *g* for 30 min. The supernatant’s amount was then calculated. The amount of water that could be absorbed by 1 gram of wheat was measured as mL as water absorption capacity.

### 2.9. Swelling Volume and Solubility Measurement of Flours

Swelling volume and solubility were determined according to Marta et al. [9]. The sample was suspended in 12.5 mL of water at a concentration of 0.35 g (db) and then mixed using a vortex for 30 s. The samples were centrifuged using a “Beckman Model TJ-6 Centrifuge” at 3500× *g* (25 °C, 30 min) after being kept in a water bath at 90 °C for 20 min and cooled in cold water. The volume of the supernatant was measured to determine the swelling volume result and then dried in an oven at 110 °C for 24 h to measure the solubility of the samples.

### 2.10. Texture Evaluation (TPA) of Pancakes

Texture evaluation was measured using a Texture Profile Analyzer (TA.XTExpress, Stable Micro System, Godalming, ENG, UK) and exponent lite express software for data collection and calculation. The product was pressed using an aluminum cylinder probe P36R with a 2 kg load cell at a speed of 5 mm/s to a strain of 50%. The product texture profile including hardness, adhesiveness, springiness, cohesiveness, chewiness, and resilience was determined from exponent lite express software.

### 2.11. Hedonic Sensory Test of Pancakes

A hedonic sensory test of the pancake was conducted with 20 panelists aged between 18 and 25 years old, both male and female. Pancakes were presented as whole pieces and placed on white plastic dishes coded with random three-digit numbers. Pancakes were evaluated based on the acceptability of their color, aroma, texture, taste, and overall appearance using a 5-point hedonic scale. The scale ranged from “extremely like” to “extremely dislike,” corresponding to the highest and lowest scores of “5” and “1”, respectively [30].

### 2.12. Statistical Analysis

Data were analyzed using one-way ANOVA, then followed by Duncan Test to detect differences. Significance was confirmed at *p* values < 0.05.

## 3. Results

### 3.1. Proximate and Crude Fiber Composition

The proximate and crude fiber test results for wheat flour, mocaf flour, arrowroot tuber flour, suweg flour, and composite flour, respectively, are presented in Table 2.

No composite flours were significantly different from each other in the parameters of moisture content, ash content, fat content, protein content, and crude fiber content, but all composite flours were significantly different from wheat flour (Table 2). The composite flour had a moisture content ranging from 10.05% to 10.87%, an ash content ranging from 2.40% db to 2.45% db, a fat content ranging from 0.36% db to 0.56% db, a protein content ranging from 1.74% db to 1.81% db, and a crude fiber content ranging from 1.38% db to 1.50% db. All composite flours had a lower moisture content, fat content, and protein content compared to wheat flour, but had a higher ash content and crude fiber content compared to wheat flour.

### 3.2. Pasting Properties

Amylographic tests were carried out to show the pasting properties of the starch paste and flour produced as well as changes in starch viscosity during the cooking process, as these characteristics play an important role in flour applications in the food industry. The pasting properties of wheat flour, mocaf flour, arrowroot tuber flour, suweg flour, CF1, CF2, and CF3 are presented in Figure 1 and tabulated in Table 3.

All parameters of pasting properties of composite flours were significantly different from wheat flour (Table 3). Pasting temperature CF1 was significant difference from CF3, where the pasting temperature increases with the increase in suweg flour (84.07 °C). Peak viscosity between samples was significantly different, where mocaf flour had the highest peak viscosity (3351.33 cP) and CF1 had the lowest (2394.67 cP). The peak viscosity of composite flour was influenced by the ratio of the addition of arrowroot flour; the higher the ratio of arrowroot flour, the higher the peak viscosity. This was due to arrowroot flour having a higher peak viscosity (2851.33 cP) when compared to suweg flour (2798.67 cP). Wheat flour has the lowest hold viscosity (935.67 cP) and breakdown viscosity (783.00 cP). The breakdown viscosity of composite flour was influenced by the ratio of arrowroot flour, where arrowroot flour has a higher breakdown viscosity (1483.00 cP) when compared to suweg flour (1093.33 cP). However, the addition of 25% of arrowroot flour can give a significant effect on breakdown viscosity, whereas the addition of 15% and 20% of arrowroot flour did not provide a significant difference. In contrast to breakdown viscosity, setback viscosity in composite flour was influenced by the ratio of suweg flour. This was because the setback viscosity of suweg flour has a significantly higher value (995.67 cP) when compared to arrowroot flour (436.00 cP). However, the addition of suweg flour only contributed significantly after the addition of 15%, whereas the addition of 10% and 5% did not have a significant effect on setback viscosity.

### 3.3. Color Evaluation of Flour

Color is one of the main factors that determine consumer acceptance [31]. The color of flour was influenced by several factors, such as flour preparation, and the presence of macronutrients, such as lipids and proteins [32]. The color of the flours is shown in Figure 2. The color of the flour was obtained using a CM-5 Spectrophotometer.

Testing the color characteristics of the flours was carried out with the CM-5 Spectrophotometer which will produce L, a*, and b* data, defined by CIE (Commission International de I’Exlairage) and tabulated in Table 4.

CF3 has the highest L* (91.19), indicating a higher lightness, whereas CF1 has the lowest L* (89.58) compared to other composite flours. Similar results were also found in the whiteness index, whereas CF1 (70:15:15) has the lowest whiteness (86.16), while CF3 (70:25:5) has the highest whiteness index (87.27) when compared to other composite flours. Whiteness index indicates the degree to which a surface was white. However, when compared to all samples, suweg flour has the lowest lightness (81.64) and whiteness index (77.46), while mocaf flour shows the highest lightness (94.17) and whiteness index (88.63). The dark color of the suweg flour causes the suweg flour to have the largest ∆E (11.94) compared to all samples, which shows a very large difference when compared to wheat flour as a control.

CF1 (70:15:15) has the highest a* (1.06), which indicates a more reddish color, while CF3 (70:25:5) has the lowest a* (0.70) when compared to all composite flours. All composite flours were significantly different from one another, and significantly different from wheat flour, mocaf flour, arrowroot flour, and suweg flour. In contrast to a*, it was known that all composite flours were not significantly different in b*. All composite flours have a higher level of red color, but a lower level of yellow color compared to wheat flour.

### 3.4. Functional Properties

Functional properties are physicochemical properties that affect the behavior of components during the process of preparation, processing, storage, and consumption. The functional properties observed in this study include water absorption capacity (WAC), swelling volume, and solubility, which are presented in Table 5.

The water absorption capacity (WAC) of the composite flour ranges from 1.62 g/g to 1.65 g/g, swelling volume ranges from 21.01 mL/g to 25.33 mL/g, and solubility ranges from 12.86% to 16.20%. The WAC of CF1, CF2, and CF3 were not significantly different from each other, but were significantly different from wheat flour (1.11 g/g). The swelling volume of all composite flours was significantly different from wheat flour. The WAC and swelling volume of the composite flours were significantly higher than wheat flour. 

### 3.5. Texture Evaluation of Pancake

The texture is one important factor that affects the final product made from the resulting flour. The characteristics of the texture can be tested using the Texture Profile Analyzer (TPA) with the parameters of hardness, springiness, cohesiveness, adhesiveness, resilience, and chewiness (Table 6).

Compared to all gluten-free pancake products, P1 had the highest hardness and chewiness and was not significantly different from pancakes made from wheat flour. Apart from hardness and chewiness, P1 was also not significantly different compared to pancakes made from wheat flour in adhesiveness. All pancake samples showed no significant differences in cohesiveness and resilience. Overall, the P1 pancake product had the most texture characteristics resembling wheat flour pancakes.

### 3.6. Color Evaluation of Pancakes

Color is the main attribute of the appearance of food products and is an important characteristic of their quality. Color is one of the determining indicators for consumers in purchasing a food product because it is a parameter that can be assessed directly before buying a product. The color of all pancake samples can be seen in Figure 3 and tabulated in Table 7.

P3 has the highest L* (49.52), which indicates a higher level of brightness, while P1 has the lowest L* (43.70) compared to all gluten-free pancakes. However, the pancake made from wheat flour still had the highest L* (52.20) when compared to all pancake products. All gluten-free pancake products have a* ranging from 9.47 to 13.74, whereas all gluten-free pancake products have a* that were significantly different from one another. P2 had the highest a* (13.74) compared to other gluten-free pancake products, which had a more reddish color and was not significantly different from wheat flour, while P3 (70:25:5) has the lowest b* (9.47). All gluten-free pancake products had lower red and yellow color levels compared to wheat flour.

### 3.7. Hedonic Sensory

Analysis of the organoleptic characteristics of pancake products was carried out using a hedonic test to determine the panelists’ preference and acceptance of the products. The parameters in this hedonic test include color, aroma, texture, taste, and overall appearance. The scoring for this hedonic test was 1 (dislike very much), 2 (dislike), 3 (normal), 4 (like), and 5 (very like). The hedonic test was carried out with 20 panelists. The results of the panelists’ preference for pancake products can be seen in Figure 4.

Pancakes made from wheat flour still had the preferred color, texture, flavor, and overall appearance. In terms of aroma, P1 had the most preferred aroma by the panelists. Overall, all pancake products have a score above 3, which indicates the product was acceptable to the panelists.

## 4. Discussion

Moisture content of the composite flours were found to be below the maximum required level (<14% *w*/*w*), so they were safe enough to prevent the growth of mold [31]. The ash content of composite flour was from 2.40% to 2.45%. The research results showed that arrowroot flour had the highest ash content (5.34%) compared to all other samples, which was in line with Sudaryati et al. [33]. All composite flours were not significantly different from each other in the parameters of moisture content, ash content, fat content, protein content, and crude fiber content, which shows that the addition of a ratio of 5%, 10%, and 15% in arrowroot tuber flour and suweg flour does not have a significant effect on any of the composite flours.

The pasting properties are an important parameter to determine the characteristics of flour. All the composite flours had a higher pasting temperature compared to wheat flour. A higher pasting temperature of composite flours indicates a higher resistance to swelling and rupture [34]. Regarding peak viscosity, all samples differed significantly from one another. A high peak viscosity might be caused by a low amylose content, which encourages swelling of starch molecules and causes an increase in viscosity [35]. Peak viscosity correlates with the quality of the final product produced, where a high peak viscosity will give a good paste texture [36]. All composite flours also showed a higher breakdown viscosity compared to wheat flour. Inversely correlated with hold viscosity and breakdown viscosity, composite flours have a lower final viscosity and setback viscosity. These results were in accordance with research conducted by Yulianti et al. [37], where the final viscosity of gluten-free composite flours shows lower results when compared to wheat flour. All composite flours have significantly lower setback than wheat flour. A lower setback for composite flours indicates a lower retrogradation tendency, which will affect the hardness of the resulting pancake product [38]. However, the results showed that the hardness of pancakes was not significantly different between pancakes made with wheat flour or with P1 or P2.

Color is also an important parameter to determine flour quality. The concentration of the suweg flour affected the color of all composite flours, and the lightness gradually decreased when the concentration of the suweg flour (81.64) increased. Previous research has shown that suweg flour has a high phenolic content (17.66 mg GAE/g) [39], while arrowroot flour only has a phenolic content of 0.268 mg GAE/g [40]. Phenolics are compounds that are easily oxidized, which can cause discoloration due to the formation of oxidation products. The presence of high phenolic content in suweg flour causes the color to brown more quickly. In line with the level of lightness, the whiteness of all samples and all composite flours were significantly different from each other and decreased as the ratio of suweg flour increased because the suweg flour had the lowest whiteness when compared to all samples (77.46).

Functional properties testing was also carried out to determine the properties of the flour to be used in making pancakes. The WAC of the composite flours (CF1, CF2, and CF3) were higher than wheat flour. High water binding capacity will increase the reconstitution ability and textural properties of the dough obtained [41]. This was in accordance with research conducted by Chandra and Samsher [42], where gluten-free flour has a higher WAC when compared to wheat flour. A high WAC indicates that flour can used in the formulation of several foods, such as dough and bakery products [43]. Suweg flour has the highest WAC (2.01 g/g) compared to all samples. This was related to the high crude fiber content in suweg flour, where the fiber-rich dough causes higher water absorption [44]. Functional properties have a correlation with amylographic properties. When compared with amylographic properties, swelling volume was directly proportional to the peak viscosity [38]. This was consistent with the results of this study, where wheat flour has the lowest peak viscosity and swelling volume compared to all samples.

In pancake products, texture is the most important parameter to determine its quality. It was known that all gluten-free pancake products have significantly different hardness. Of all gluten-free pancakes, the hardness of P1 and P2 were not significantly different from wheat flour pancakes. This shows that P1 and P2 have good texture characteristics in terms of hardness and resemble pancakes made from wheat flour. Hardness was affected by the difference in the ratio of suweg flour in the composite flour. The hardness of the product increases as more suweg flour is added to the composite flour, indicating that the suweg flour affects the hardness of the resulting pancake product. This could be due to the high content of crude fiber in suweg flour causing the hardness of the product to increase. In pancake products, springiness was the desired texture attribute [45]. Springiness was associated with the freshness of the product produced, where high quality products have a higher springiness [46]. It was known that P2 and P3 were not significantly different from wheat flour pancakes. Meanwhile P1 had a significantly lower springiness compared to pancake products made from wheat flour. Suweg flour contains resistant starch, which can increase the viscosity of the dough and the resulting final product. In addition, suweg flour also has a high crude fiber content, which can give a chewy texture to the resulting product. Based on the research results, it was known that P2 and P3 had a lower chewiness when compared to wheat flour pancake products. In contrast to springiness, an increase in chewiness was undesirable for pancake products. This was because the increased chewiness made the final pancake product to have a texture that was too chewy rather than soft [45]. P1 had a chewiness that was not significantly different from pancakes made from wheat flour, which shows good characteristics and resembles wheat flour.

Color is one of the determining indicators for consumers in purchasing a food product, so it is one of the criteria that need to be considered when determining the acceptability of the resulting pancake product. Lightness was affected by the ratio of flour used. The higher the ratio of arrowroot flour added, the lighter the product that is produced. This was consistent with research by Lestari et al. [47], where the addition of the highest arrowroot flour ratio (30%) resulted in cookies with the highest brightness. Flour color concentration had the same result; the concentration of suweg flour affected the color of the pancake product, and the L* decreases when suweg flour concentration was higher. This was due to the basic color of the suweg flour (dark brown), which affects the final product [48]. All gluten-free pancakes have lower red and yellow color levels when compared to wheat flour. The presence of red and yellow colors in flour can be influenced by the presence of chemical compositions such as fats and proteins [32]. This was consistent with the data obtained, where all composite flour has lower fat and protein compared to wheat flour. Calculations were also made for the whiteness of pancake products to determine the level of whiteness of the final product. P3 (70:25:5) had the highest whiteness (40.54), and P1 (70:15:15) had the lowest whiteness (34.36). Similar to the lightness, the whiteness was affected by the ratio of the addition of suweg flour, where the higher the ratio of suweg flour added, the lower the whiteness. In the ∆E calculation, P1 had the highest ∆E compared to other gluten-free pancake products, which indicates a greater color difference when compared to pancake products made from wheat flour. Meanwhile, P2 has the smallest ∆E, which shows less difference compared to wheat flour pancakes.

In terms of panelist preferences, results showed that all pancake products were acceptable to panelists. In previous research, the addition of suweg flour produced a taste that was not liked by the panelists [48]. The results of this study show that mixing several types of flour gives good results and can complement the deficiencies of individual flours. The textural preference for gluten-free pancake products was influenced by the addition of arrowroot flour ratio, where P1 had the lowest ratio of arrowroot flour and was not significantly different when compared to pancake products made from wheat flour, whereas P2 and P3 had significant differences. The addition of an arrowroot flour ratio of 15% gives the pancake products a characteristic of firm texture but remains smooth when consumed. This was a positive result as, in previous studies, the resulting gluten-free pancakes had unfavorable characteristics, such as a crumbly texture [16]. P1 was the most preferred among the gluten-free pancakes es and had a more favorable aroma than wheat flour-based pancakes.

## 5. Conclusions

Based on the results, all pancake products were acceptable to panelists, indicating that they had good overall characteristics which resembled a pancake made from wheat flour. The existence of positive results in the manufacture of gluten-free composite flour means that it could be an alternative that can replace wheat flour. The results of this research can also be further developed to obtain gluten free composite flour that increasingly resembles wheat flour. The ratio of mocaf flour, arrowroot flour, and suweg flour had a significant effect the characteristics of the flour, such as pasting temperature, peak viscosity, hold viscosity, breakdown viscosity, setback, L*, a*, hue, whiteness, ∆E, as well as swelling volume and solubility. The ratio of mocaf flour, arrowroot flour, and suweg flour also had a significant effect on the texture characteristics of hardness, adhesiveness, chewiness, color characteristics L*, a*, whiteness, ∆E, and preference for the aroma of the resulting gluten-free pancake product. The best formulation for the resulting pancake product was P1, which was composed of 70% mocaf flour, 15% arrowroot tuber flour, and 15% suweg flour with characteristics and organoleptics hardness texture 2698.16, springiness texture 0.98, chewiness texture 2199.06, L* 43.70, a* 12.11, b* 31.41, organoleptic taste 3.76, organoleptic color 3.33, organoleptic texture 3.71, aroma organoleptic 3.67, and overall appearance organoleptic 3.62.

## Figures and Tables

**Figure 1 foods-12-01892-f001:**
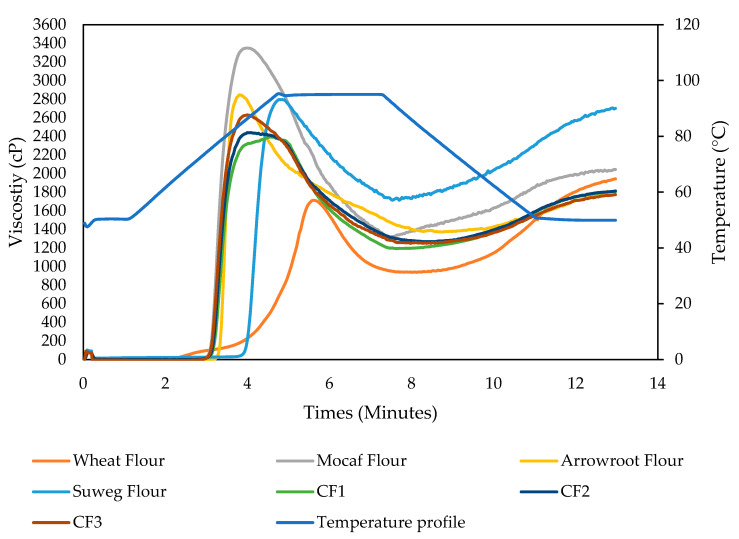
Pasting properties of wheat flour, mocaf flour, arrowroot flour, suweg flour, and composite flours CF1, CF2, CF3.

**Figure 2 foods-12-01892-f002:**
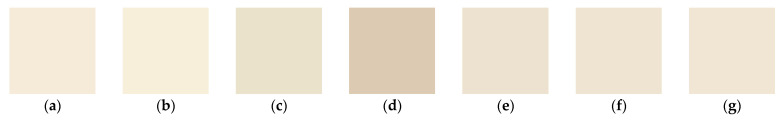
Visual appearance of wheat flour (**a**), mocaf flour (**b**), arrowroot tuber flour (**c**), suweg flour (**d**), and composite flour CF1 (**e**), CF2 (**f**), and CF3 (**g**).

**Figure 3 foods-12-01892-f003:**
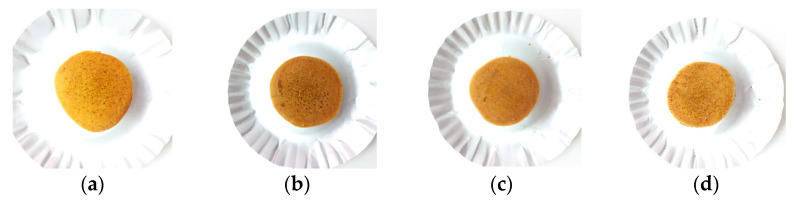
The visual appearance of pancakes with wheat flour (**a**), with 70% mocaf flour, 15% arrowroot tuber flour, and 15% suweg flour (**b**), with 70% mocaf flour, 20% arrowroot tuber flour, and 10% suweg flour (**c**), with 70% mocaf flour, 25% arrowroot tuber flour, and 5% suweg flour (**d**).

**Figure 4 foods-12-01892-f004:**
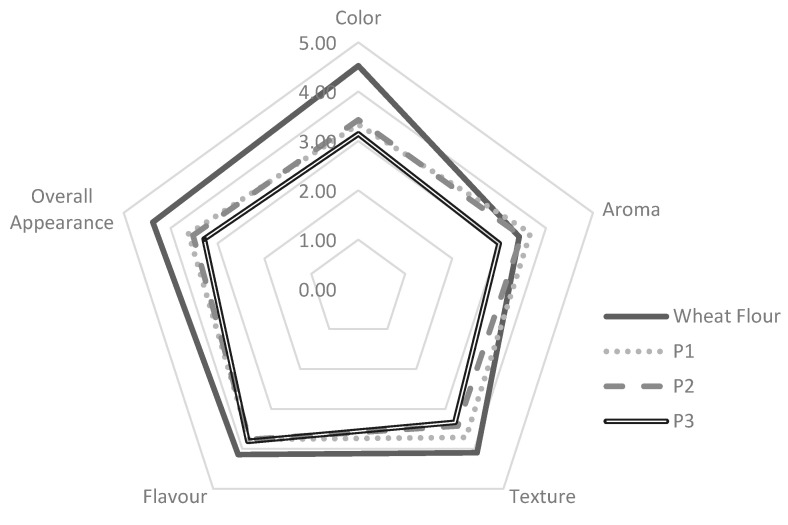
Organoleptic parameters of pancakes.

**Table 1 foods-12-01892-t001:** Pancake formulations.

Formulation	Mocaf (g)	Arrowroot Tuber (g)	Suweg (g)	Powdered Sugar (g)	Powdered Milk (g)	Baking Powder (g)	Egg (pcs)	Water (mL)
Pancake 1 (P1)	70	15	15	90	26	9	1	170
Pancake 2 (P2)	70	20	10	90	26	9	1	170
Pancake 3 (P3)	70	25	5	90	26	9	1	170

**Table 2 foods-12-01892-t002:** Proximate composition of wheat flour, mocaf flour, arrowroot tuber flour, suweg flour, and composite flours.

Sample	Water (%db)	Ash (%db)	Fat (%db)	Protein (%db)	Crude Fiber (%db)
WF	14.52 ± 0.16 ^a^	0.69 ± 0.01 ^e^	1.36 ± 0.43 ^a^	10.55 ± 0.18 ^a^	0.38 ± 0.15 ^e^
MF	14.33 ± 0.25 ^a^	1.46 ± 0.20 ^d^	1.05 ± 0.06 ^ab^	1.13 ± 0.22 ^d^	1.18 ± 0.09 ^d^
AF	7.32 ± 0.05 ^c^	5.34 ± 0.02 ^a^	0.54 ± 0.28 ^bc^	4.40 ± 0.10 ^b^	1.79 ± 0.05 ^b^
SF	10.21 ± 0.71 ^b^	4.05 ± 0.06 ^b^	0.53 ± 0.34 ^bc^	4.44 ± 0.06 ^b^	3.15 ± 0.25 ^a^
CF1	10.87 ± 0.25 ^b^	2.45 ± 0.07 ^c^	0.56 ± 0.01 ^bc^	1.75 ± 0.07 ^c^	1.38 ± 0.01 ^cd^
CF2	10.26 ± 0.83 ^b^	2.40 ± 0.02 ^c^	0.36 ± 0.07 ^c^	1.74 ± 0.05 ^c^	1.50 ± 0.01 ^bc^
CF3	10.05 ± 0.10 ^b^	2.43 ± 0.02 ^c^	0.43 ± 0.20 ^bc^	1.81 ± 0.10 ^c^	1.44 ± 0.13 ^cd^

Means within columns with different superscripts are significantly different (*p* ≤ 0.05). CF1 = MF:AF:SF = 70:15:15, CF2 = MF:AF:SF = 70:20:10, CF3 = MF:AF:SF = 70:25:5.

**Table 3 foods-12-01892-t003:** Pasting properties of wheat flour, mocaf flour, arrowroot tuber flour, suweg flour, and composite flours.

Sample	Pasting Temperature (°C)	Peak Viscosity (cP)	Hold Viscosity (cP)	Final Viscosity (cP)	Breakdown (cP)	Setback (cP)
WF	64.91 ± 0.60 ^e^	1718.67 ± 4.93 ^g^	935.67 ± 14.50 ^e^	1943.00 ± 24.06 ^c^	783.00 ± 10.15 ^f^	1007.33 ± 9.71 ^a^
MF	73.17 ± 0.24 ^d^	3351.33 ± 17.79 ^a^	1312.00 ± 39.40 ^bc^	2042.33 ± 10.02 ^b^	2039.33 ± 56.22 ^a^	730.33 ± 34.93 ^b^
AF	76.62 ± 0.04 ^b^	2851.33 ± 29.87 ^b^	1368.67 ± 55.64 ^b^	1804.33 ± 98.54 ^d^	1483.00 ± 25.24 ^b^	436.00 ± 45.92 ^e^
SF	84.07 ± 0.15 ^a^	2798.67 ± 10.97 ^c^	1705.33 ± 55.23 ^a^	2701.00 ± 12.12 ^a^	1093.33 ± 47.26 ^e^	995.67 ± 53.93 ^a^
CF1	74.06 ± 0.47 ^c^	2394.67 ± 38.03 ^f^	1191.67 ± 14.01 ^d^	1806.67 ± 36.12 ^d^	1203.00 ± 24.02 ^d^	615.00 ± 22.11 ^c^
CF2	73.79 ± 0.10 ^cd^	2445.67 ± 15.50 ^e^	1264.00 ± 5.20 ^c^	1813.00 ± 16.46 ^d^	1181.67 ± 12.86 ^d^	549.00 ± 19.47 ^d^
CF3	73.24 ± 0.47 ^d^	2632.00 ± 24.56 ^d^	1245.67 ± 44.64 ^cd^	1772.67 ± 6.03 ^d^	1386.33 ± 39.80 ^c^	527.00 ± 33.63 ^d^

Means within columns with different superscripts are significantly different (*p* ≤ 0.05). CF1 = MF:AF:SF = 70:15:15, CF2 = MF:AF:SF = 70:20:10, CF3 = MF:AF:SF = 70:25:5.

**Table 4 foods-12-01892-t004:** Color evaluation of wheat flour, mocaf flour, arrowroot flour, suweg flour, and composite flours.

Sample	L*	a*	b*	Hue	Whiteness Index	∆E
WF	93.09 ± 0.21 ^b^	0.45 ± 0.01 ^e^	9.81 ± 0.12 ^c^	1.53 ± 0.00 ^a^	87.99 ± 0.03 ^b^	0.00 ± 0.00 ^f^
MF	94.17 ± 0.16 ^a^	0.43 ± 0.02 ^e^	9.75 ± 0.13 ^c^	1.53 ± 0.00 ^a^	88.63 ± 0.07 ^a^	1.09 ± 0.31 ^e^
AF	89.48 ± 0.21 ^e^	−0.36 ± 0.02 ^f^	11.35 ± 0.16 ^b^	−1.54 ± 0.00 ^f^	84.52 ± 0.12 ^f^	4.01 ± 0.27 ^b^
SF	81.64 ± 0.42 ^f^	1.78 ± 0.02 ^a^	12.96 ± 0.28 ^a^	1.43 ± 0.00 ^e^	77.46 ± 0.18 ^g^	11.94 ± 0.19 ^a^
CF1	89.58 ± 0.37 ^e^	1.06 ± 0.01 ^b^	9.04 ± 0.03 ^d^	1.45 ± 0.00 ^d^	86.16 ± 0.28 ^e^	3.65 ± 0.51 ^b^
CF2	90.37 ± 0.16 ^d^	0.88 ± 0.03 ^c^	9.13 ± 0.26 ^d^	1.47 ± 0.00 ^c^	86.70 ± 0.28 ^d^	2.84 ± 0.34 ^c^
CF3	91.19 ± 0.05 ^c^	0.70 ± 0.03 ^d^	9.17 ± 0.27 ^d^	1.49 ± 0.00 ^b^	87.27 ± 0.22 ^c^	2.03 ± 0.19 ^d^

Means within columns with different superscripts are significantly different (*p* ≤ 0.05). CF1 = MF:AF:SF = 70:15:15, CF2 = MF:AF:SF = 70:20:10, CF3 = MF:AF:SF = 70:25:5.

**Table 5 foods-12-01892-t005:** Functional properties of wheat flour, mocaf flour, arrowroot tuber flour, suweg flour, and composite flours.

Sample	WAC (g/g)	Swelling Volume (mL/g)	Solubility (%)
WF	1.11 ± 0.13 ^d^	11.52 ± 0.31 ^e^	14.17 ± 2.00 ^c^
MF	1.68 ± 0.01 ^b^	16.65 ± 0.24 ^d^	9.20 ± 0.36 ^d^
AF	1.48 ± 0.04 ^c^	22.19 ± 0.42 ^b^	19.05 ± 0.36 ^a^
SF	2.01 ± 0.07 ^a^	16.55 ± 0.22 ^d^	17.35 ± 0.17 ^b^
CF1	1.63 ± 0.03 ^b^	25.33 ± 0.62 ^a^	12.86 ± 0.80 ^c^
CF2	1.65 ± 0.02 ^b^	21.10 ± 0.16 ^c^	16.20 ± 0.10 ^b^
Cf3	1.62 ± 0.03 ^b^	21.01 ± 0.32 ^c^	16.11 ± 0.37 ^b^

Means within columns with different superscripts are significantly different (*p* ≤ 0.05). CF1 = MF:AF:SF = 70:15:15, CF2 = MF:AF:SF = 70:20:10, CF3 = MF:AF:SF = 70:25:5.

**Table 6 foods-12-01892-t006:** Pancake texture evaluation made from wheat flour and composite flour.

Sample	Hardness (g)	Adhesiveness (g.sec)	Springiness	Cohesiveness	Chewiness	Resilience
Wheat flour	2632.36 ± 232.56 ^ab^	−14.19 ± 2.73 ^bc^	0.99 ± 0.00 ^a^	0.85 ± 0.01 ^a^	2211.92 ± 206.55 ^a^	0.66 ± 0.04 ^a^
P1	2698.16 ± 162.34 ^a^	−19.92 ± 3.51 ^c^	0.98 ± 0.01 ^b^	0.84 ± 0.00 ^a^	2199.06 ± 112.78 ^a^	0.62 ± 0.04 ^a^
P2	2372.49 ± 27.47 ^b^	−17.24 ± 4.77 ^bc^	0.98 ± 0.01 ^ab^	0.83 ± 0.00 ^a^	1926.63 ± 14.80 ^b^	0.62 ± 0.04 ^a^
P3	2035.74 ± 64.56 ^c^	−11.42 ± 0.20 ^a^	0.98 ± 0.01 ^ab^	0.84 ± 0.00 ^a^	1677.32 ± 32.45 ^c^	0.64 ± 0.04 ^a^

Means within columns with different superscripts are significantly different (*p* ≤ 0.05). P1 = pancake formulation using composite flour 1, P2 = pancake formulation using composite flour 2, P3 = pancake formulation using composite flour 3.

**Table 7 foods-12-01892-t007:** Pancake color evaluation made from wheat flour and composite flour.

Sample	L*	a*	b*	Hue	Whiteness	∆E
Wheat Flour	52.20 ± 3.24 ^a^	14.13 ± 1.35 ^a^	37.77 ± 2.27 ^a^	1.21 ± 0.05 ^ab^	37.38 ± 1.41 ^b^	0.00 ± 0.00 ^d^
P1	43.70 ± 1.64 ^c^	12.11 ± 0.07 ^b^	31.41 ± 2.58 ^d^	1.20 ± 0.03 ^ab^	34.36 ± 0.21 ^c^	12.93 ± 1.18 ^a^
P2	47.06 ± 2.79 ^bc^	13.74 ± 0.55 ^a^	35.24 ± 1.63 ^d^	1.20 ± 0.03 ^b^	34.89 ± 1.47 ^c^	5.27 ± 0.49 ^c^
P3	49.52 ± 0.56 ^ab^	9.47 ± 0.78 ^c^	29.89 ± 2.56 ^d^	1.26 ± 0.02 ^a^	40.54 ± 1.28 ^a^	7.75 ± 0.52 ^b^

Means within columns with different superscripts are significantly different (*p* ≤ 0.05).

## Data Availability

The data presented in this study are available on request from the corresponding author. The data are not publicly available due to their containing information that could compromise the privacy of research participants.

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
