# Peer review of "Application of Composite Flour from Indonesian Local Tubers in Gluten-Free Pancakes"

_foods, 2023, doi:10.3390/foods12091892_

Round 1

Reviewer 1 Report

The submitted article presents an innovative study demonstrating that composite flour made from mocaf, arrowroot, and suweg flour can produce gluten-free pancakes with characteristics similar to wheat flour-based pancakes. The study's findings have implications for the food industry and highlight the potential for using local commodities to develop gluten-free products. The authors' work also emphasizes the importance of considering the impact of food production on people with dietary restrictions, such as gluten intolerance. There are a few areas for improvement:

The introduction requires further elaboration to provide a comprehensive background on previous research on gluten-free pancake production using alternative raw materials. 

The authors should highlight the novelty of their research by showcasing the limitations of previous studies and emphasizing the contributions of their work.

The initial and final moisture content of arrowroot flour and suweg flour used in the study should be provided to facilitate the replication of the experiment. Moreover,

the names of the producers for every ingredient used for the pancake preparation should be included to enable readers to evaluate the quality of the raw materials.

The authors should provide an overall conclusion at the end of the section to summarize their findings and their implications for the food industry.

The reference list should be expanded to include more relevant literature on the topic.

Moderate editing of the English language is required.

Author Response

Thank you very much for your comments concerning our manuscript entitled “Application of Composite Flour from Indonesian Local Tubers in Gluten-Free Pancakes”. Those comments are valuable and very helpful for revising and improving our paper. We have studied the comments carefully and have made a correction which we hope meets with approval. The revised portions are marked in yellow on the paper. The main correction and the responses to the reviewer’s comment are as follows:

The submitted article presents an innovative study demonstrating that composite flour made from mocaf, arrowroot, and suweg flour can produce gluten-free pancakes with characteristics similar to wheat flour-based pancakes. The study's findings have implications for the food industry and highlight the potential for using local commodities to develop gluten-free products. The authors' work also emphasizes the importance of considering the impact of food production on people with dietary restrictions, such as gluten intolerance. There are a few areas for improvement:

  1. The introduction requires further elaboration to provide a comprehensive background on previous research on gluten-free pancake production using alternative raw materials. 

          Response:

          It has been revised (Lines 67-68; 75-87)

  1. The authors should highlight the novelty of their research by showcasing the limitations of previous studies and emphasizing the contributions of their work.

Response:

It has been revised (Lines 75-87; 426-428; 434-436)

  1. The initial and final moisture content of arrowroot flour and suweg flour used in the study should be provided to facilitate the replication of the experiment. Moreover,

Response:

It has been presented in Table 1

  1. The names of the producers for every ingredient used for the pancake preparation should be included to enable readers to evaluate the quality of the raw materials.

          Response:

It has been revised (Lines 91-93; 95; 120-122)

  1. The authors should provide an overall conclusion at the end of the section to summarize their findings and their implications for the food industry.

          Response:

It has been revised (Lines 439-443)

  1. The reference list should be expanded to include more relevant literature on the topic.

          Response:

It has been added. We have added 14 relevant kinds of literature (marked in yellow in the reference list, Ref. 13-15; Ref. 17-27)

Reviewer 2 Report

line 67: pancakes are fried, not baked

section 2.4: the same informations are in table 1, remove this section

line 119: what is aquadest ?

figure 1: preferably use colored lines or lines with markers or different types of lines

figure 2: how were these pictures obtained ?

table 4: what is whiteness, please define

table 6: specify units (where applicable)

line 292: wheat flour or pancake made from wheat flour, be precise

Results, Discussion: separation of results and discussion makes the discussion largely a repetition of the description of results, e.g. lines 292-293 and 394-396: However, wheat flour still had the highest L* (52.20) when compared to all pancake products. However, pancake made from wheat flour still had the highest L* (52.20) when compared to all pancake products. Please rewrite the discussion, excluding the re-description of the results, focusing on explaining the observations, dependencies, effects described in the Results.

English needs improvement. American (e.g. fiber) and British (e.g. colour) spellings are mixed up. The past tense should be used in the research description, e.g. line 84: The arrowroot tubers were washed and peeled instead of The arrowroot tubers are washed and peeled. Likewise lines 88 (is), 93 (are) and others.

Author Response

Response to the reviewers’ comments:

Reviewer 2

Thank you very much for your comments concerning our manuscript entitled “Application of Composite Flour from Indonesian Local Tubers in Gluten-Free Pancakes”. Those comments are valuable and very helpful for revising and improving our paper. We have studied the comments carefully and have made a correction which we hope meets with approval. The revised portions are marked in yellow on the paper. The main correction and the responses to the reviewer’s comment are as follows:

  1. Line 67: pancakes are fried, not baked

          Response:

          It has been revised (Lines 67)

  1. section 2.4: the same informations are in table 1, remove this section

          Response:

          It has been removed (Lines 114)

  1. line 119: what is aquadest?

          Response:

          It has been revised (Lines 133-134)

  1. figure 1: preferably use colored lines or lines with markers or different types of lines

          Response:

          It has been revised (Fig 1, line 203)

  1. figure 2: how were these pictures obtained?

          Response:

          It has been revised (Lines 234-235)

  1. table 4: what is whiteness, please define

          Response:

It has been revised to whiteness index. Whiteness index indicates the degree to which a surface was white (line 251)

  1. table 6: specify units (where applicable)

          Response:

          It has been revised (Table 6, line 285)

  1. line 292: wheat flour or pancake made from wheat flour, be precise

Response:

It has been revised (Lines 308-309)

  1. Results, Discussion: separation of results and discussion makes the discussion largely a repetition of the description of results, e.g. lines 292-293 and 394-396: However, wheat flour still had the highest L* (52.20) when compared to all pancake products. However, pancake made from wheat flour still had the highest L* (52.20) when compared to all pancake products. Please rewrite the discussion, excluding the re-description of the results, focusing on explaining the observations, dependencies, effects described in the Results.

Response:

It has been revised. the re-description of the results has been excluded (result and discussion, lines 403-407)
